# Pd/NiMoO₄/NF electrocatalysts for the efficient and ultra-stable synthesis and electrolyte-assisted extraction of glycolate

Kai Shi[1], Di Si[1], Xue Teng[1], Lisong Chen [1,2] ✉ & Jianlin Shi [3]

Electrocatalytic conversion of organic small molecules is a promising technique for value-added chemical productions but suffers from high precious metal consumption, poor stability of electrocatalysts and tedious product separation. Here, a Pd/NiMoO₄/NF electrocatalyst with much lowered Pd loading amount (3.5 wt.%) has been developed for efficient, economic, and ultra-stable glycolate synthesis, which shows high Faradaic efficiency (98.9%), yield (98.8%), and ultrahigh stability (1500 h) towards electrocatalytic ethylene glycol oxidation. Moreover, the obtained glycolic acid has been converted to value-added sodium glycolate by in-situ acid-base reaction in the NaOH electrolyte, which is atomic efficient and needs no additional acid addition for product separation. Moreover, the weak adsorption of sodium glycolate on the catalyst surface plays a significant role in avoiding excessive oxidation and achieving high selectivity. This work may provide instructions for the electrocatalyst design as well as product separation for the electrocatalytic conversions of alcohols.

Sodium glycolate, an essential commodity chemical, is widely employed as an important intermediate in the production techniques of various important chemicals. The global production of sodium glycolate reaches around 0.1 million tons per year and the market price of sodium glycolate is around 3.0 thousand US dollars[1]. At present, sodium glycolate is produced through a highly energy-intensive process under high temperatures. Typically, chloroacetic acid and sodium hydroxide undergo a substitution reaction to produce sodium glycolate and sodium chloride followed by a complex separation process to obtain the final product. In addition to the high cost and energy consumption of above technology, the raw chloroacetic is usually acid-poisonous and corrosive, which is detrimental to the environment and human health (Fig. 1a). In this regard, developing efficient, energy-saving, non-toxic and harmless ways under ambient conditions to produce sodium glycolate is of great importance[2].

With the rapid development of renewable electricity generation approaches, electrosynthesis is becoming popular as an effective synthesis tool under ambient conditions[3–9]. For example, the electrochemical oxidations of organic small molecules (methanol, ethanol, ethylene glycol, glycerol, 5 hydroxymethyl furfural, and so on) to valuable chemicals such as formic acid, glycolic acid, and 2,5-furan dicarboxylic acid, have been reported[10–18]. Previously, several noble-metal electrodes have been explored for ethylene glycol oxidation; however, only C1 molecules, such as CO₂ and formate, have been obtained as the main products in most reports, and the scarcity and high cost of noble metals have also limited their large-scale application[11,19–25]. Recently, researchers load the noble metal on oxides or hydroxide substrates to obtain C2 products. For example, our group has reported the electrocatalytic oxidation of ethylene glycol (EG) to glycolic acid (GA) by PdAg/NF, which has attracted wide attention[26]. Although many researchers have made contributions to this field, there are still problems remaining[27,28]. On the one hand, the high loading amount of precious metal in the electrocatalysts and its rapid deactivation (no longer than 200 h) can't meet the requirements

[1]State Key Laboratory of Petroleum Molecular & Process engineering, Shanghai Key Laboratory of Green Chemistry and Chemical Processes, School of Chemistry and Molecular Engineering, East China Normal University, Shanghai 200062, China. [2]Institute of Eco-Chongming, Shanghai 202162, China. [3]Shanghai Institute of Ceramics, Chinese Academy of Sciences, Shanghai 200050, P. R. China. ✉e-mail: lschen@chem.ecnu.edu.cn

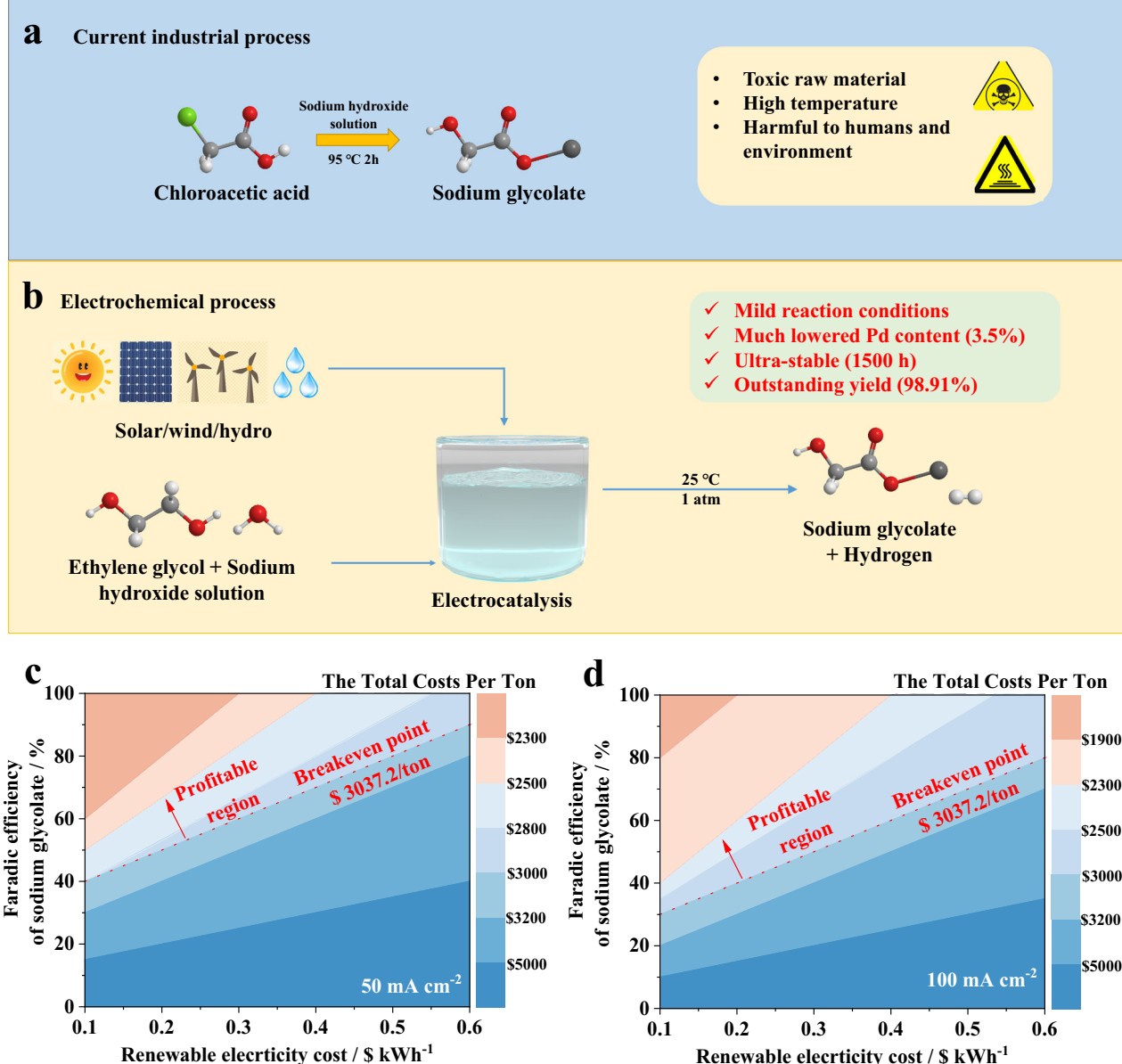

**Fig. 1 | Schematic illustration of the production of sodium glycolate. a** Current industrial route. **b** Proposed electrochemical route. **c** TEA results at 50 mA cm⁻². **d** TEA results at 100 mA cm⁻².

of large-scale industrial applications; on the other hand, excess of high concentration potassium hydroxide solution is used as the electrolyte to keep the high reaction selectivity, which need to be neutralized before product separation, therefore a large amount of alkaline and acid are inevitably wasted.

In order to solve the above problems, we propose a strategy of in-situ acid-base reaction for sodium glycolate extraction from the electrolyte. Unlike the situations in previous reports, in the electrolytic process of this report, sodium glycolate can be obtained from the in-situ reaction between the sodium hydroxide electrolyte and GA produced from EG oxidation (Fig. 1b), which not only can make full use of electrolytes to improve the utilization rate of atoms, but also minimize the waste of acid.

To demonstrate the economic potential of this approach, a techno-economic analysis (TEA) has been provided (Supplementary Note 1 and Supplementary Fig. 1)[4,29]. Preliminary TEA (Fig. 1c, d and Supplementary Fig. 2) indicates that the selective oxidation of ethylene

glycol to sodium glycolate (>80% selectivity) at a rather high current density (>100 mA cm⁻²) is profitable.

In this work, a Pd/NiMoO₄/NF electrocatalyst with much lowered Pd content (3.5 wt.%) has been developed for efficient, economic, and ultra-stable glycolate synthesis. Pd/NiMoO₄/NF shows reasonably high Faradaic efficiency (98.9%) and yield (98.8%), and ultrahigh stability (1500 h) towards electrocatalytic ethylene glycol oxidation. Moreover, the obtained glycolic acid has been converted to value-added sodium glycolate by in-situ acid-base reaction with the NaOH electrolyte, which is atomically efficient and no additional acid addition is needed for product separation. The experiment results reveal that i) the weak adsorption of sodium glycolate on the catalyst surface plays a significant role in avoiding excessive oxidation and achieving high selectivity; ii) the electron transfer between Pd and NiMoO₄ and the downshift in the d band-center of Pd promote the adsorption of OH* and the enhancement of catalytic activity and stability.

## Results

### Synthesis and characterization of catalysts

Pd/NiMoO$_4$/NF was synthesized by a two-step process as shown in Fig. 2a (Experimental details can be found in the methods part in SI). In the first step, bi-metal hydroxide was grown on nickel foam by a hydrothermal method, followed by Ar annealing to obtain the crystallized NiMoO$_4$ nanorods supported on nickel foam (NiMoO$_4$/NF)[30]. In the second step, palladium nanosheets were deposited on the NiMoO$_4$/NF by a NaBH$_4$ reduction approach to obtain the final catalyst Pd/NiMoO$_4$/NF. Control samples NiMoO$_4$/NF and Pd/NF have also been obtained by similar methods.

Firstly, X-ray diffraction (XRD) patterns of Pd/NiMoO$_4$/NF as well as other catalysts have been obtained to demonstrate the successful synthesis of these electrocatalysts. As shown in Fig. 2b, only characteristic diffraction peaks belonging to the distinct face-centered cubic lattice of Pd and the tetragonal structure of NiMoO$_4$, respectively, without other diffraction peaks being found[31,32]. Typically, the reflections located at 40.12°, 46.66° and 68.12° correspond to the (111),

(200), and (220) planes of Pd (JCPDS no. 46–1043), while the those located at 14.3°, 25.3°, 28.8°, and 32.6° correspond to the (110), ($\bar{1}$12), (220) and (022) planes of NiMoO$_4$ (JCPDS no. 33–0948), indicating the successful synthesis of the Pd/NiMoO$_4$/NF, NiMoO$_4$/NF, and Pd/NF. The microstructures of these obtained electrocatalysts were verified by scanning electron microscopy (SEM). Typical SEM images confirm that well-aligned Pd nanosheet arrays have been grown successfully on the surface of the NiMoO$_4$ nanorods (Fig. 2c, d). The SEM images of the as-obtained NiMoO$_4$/NF, Pd/NF, and NF are displayed in Supplementary Figs. 3–5. In more detail, the structure of Pd/NiMoO$_4$/NF was further confirmed by the transmission electron microscopic (TEM) imaging (Fig. 2e). By highlighting the distinct lattice fringes of 2.24 Å and 2.72 Å in the lattice d-spacing, which correspond to the (111) plane of the Pd and the ($\bar{3}$12) plane of the NiMoO$_4$ respectively, the high-resolution TEM (HRTEM) image further confirms the crystalline Pd/NiMoO$_4$/NF (Fig. 2f, g). The TEM images of the as-obtained NiMoO$_4$/NF and Pd/NF are displayed in Supplementary Fig. 6. The energy-dispersive spectroscopy (EDS) elemental mapping analysis indicates

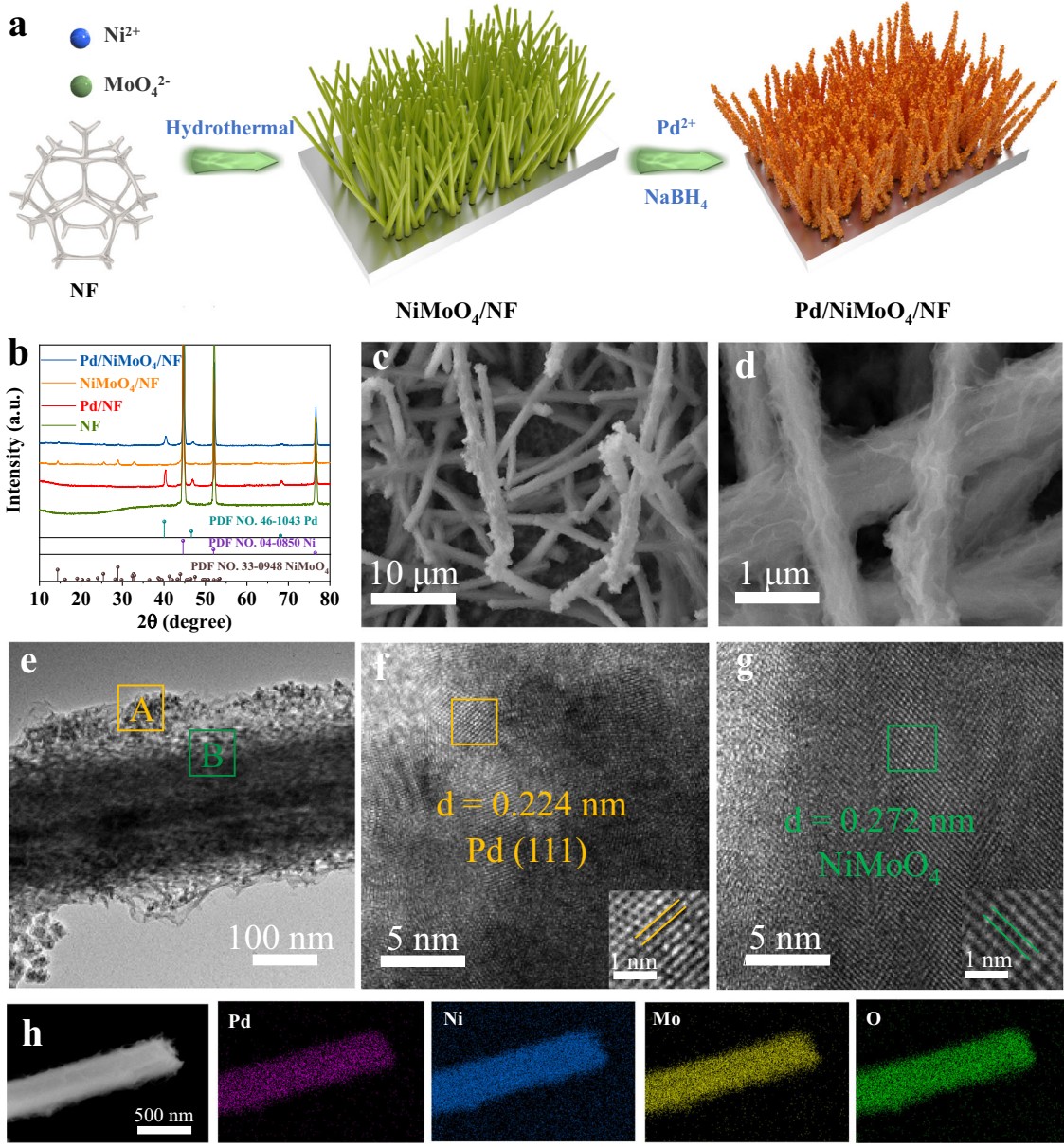

**Fig. 2 | Morphology and structures characterizations of catalysts. a** Schematic illustration for the synthesis of Pd/NiMoO$_4$/NF. **b** XRD patterns of Pd/NiMoO$_4$/NF, NiMoO$_4$/NF, Pd/NF, and pure NF. **c, d** SEM images of Pd/NiMoO$_4$/NF. **e** TEM and **f, g** HRTEM images of Pd/NiMoO$_4$/NF. **h** SEM-EDS mapping of Pd/NiMoO$_4$/NF.

that Pd, Ni, Mo, and O are homogeneously distributed throughout the Pd/NiMoO₄/NF (Fig. 2h). In addition, the precise Pd loading amount has been analyzed by inductively coupled plasma mission spectroscopy (ICP-OES), and the weight percentage of Pd is as low as 3.5 wt.%, which is much lower than those reported so far (Supplementary Fig. 7).

## Structure and electronic properties of catalysts

To further probe the chemical structure of Pd/NiMoO₄/NF, Raman and Fourier transform infrared (FT-IR) spectroscopic measurements were carried out. As shown in Fig. 3a, the bands located at 907.9 cm⁻¹ and 957.1 cm⁻¹ are attributed to the symmetric and asymmetric stretches of terminal Mo=O, while the band at 704.0 cm⁻¹ is assigned to the stretching mode of the Ni-O-Mo bond. Besides, the bands at 382.1 cm⁻¹ belongs to the bending mode of Mo-O[32]. Similar information can also be obtained from the FT-IR curves (Supplementary Fig. 8), from which the obvious characteristic peaks at 960 cm⁻¹ and 713 cm⁻¹ can be attributed to the Mo=O symmetric stretching and the Ni-Mo-O symmetrically stretching, respectively. Besides, the characteristic peaks at 450 cm⁻¹ and 412 cm⁻¹ belongs to the superposition of stretching vibration of MoO₆ and NiO₆ groups[33]. Obviously, the red shift of each peak can be observed in Raman and FT-IR spectroscopic curves, which indicates the successful loading of Pd on NiMoO₄.

Furthermore, X-ray photoelectron spectroscopy (XPS) was adopted to analyze the electronic states of Pd and the surface element composition of the Pd/NiMoO₄/NF. The survey spectrum of Pd/NiMoO₄/NF (Supplementary Fig. 9) confirms the presence of Pd, Ni, Mo, and O, which agrees well with the EDS results. As shown in Fig. 3b, the high-resolution Pd 3d spectrum of Pd/NiMoO₄/NF consists of four peaks. The peaks located at about 335.4 eV (Pd 3d₅/₂) and 340.5 eV (Pd 3d₃/₂) belongs to the metallic Pd; while the peaks located at about 337.5 eV (Pd 3d₅/₂) and 342.3 eV (Pd 3d₃/₂) corresponds to the PdO[6]. The peak of Pd 3d in Pd/NiMoO₄/NF shifts in a positive direction by 0.9 eV compared to that of the Pd/NF, implying electric interactions between Pd and NiMoO₄. Six peaks can be obviously seen in the high-resolution Ni 2p XPS spectrum (Fig. 3c), among which peaks at 855.8 and 873.6 eV are assigned to Ni 2p₃/₂ and Ni 2p₁/₂ of Ni²⁺, respectively, while peaks at 857.3 and 875.5 eV are attributed to Ni 2p₃/₂ and Ni 2p₁/₂

of Ni³⁺, respectively, and those at 862.4 and 880.8 eV are ascribed to two accompanying satellites. The NiMoO₄/NF exhibits a high proportion of Ni³⁺, which is associated with surface hydroxylation[34]. For high-resolution Mo 3d XPS spectra, three sets of doublet peaks are observed between 230 and 240 eV (Fig. 3d), which can be attributed to the doublet peaks of Mo⁶⁺ 3d₅/₂, 3d₃/₂, Mo⁵⁺3d₅/₂, 3d₃/₂, and their satellite peaks[35]. The binding energy of Ni 2p and Mo 3d for Pd/NiMoO₄/NF is negatively shifted (-0.2 eV) compared with those of NiMoO₄/NF. Besides, as shown in Supplementary Fig. 10, the charge density difference results indicate that 1.63 electrons transfer from the Pd to the NiMoO₄ substrate, which agree well the results of the XPS. The above results reveal the transferring of electrons from Pd to NiMoO₄, which further confirms the strong electronic interaction between Pd and NiMoO₄.

In order to further explore the effect of interfacial interaction on the binding strength of absorbates, the surface valence band photoemission spectroscopy was employed to evaluate the d-band center of Pd/NF and Pd/NiMoO₄/NF. As shown in Fig. 3e, f and Supplementary Fig. 11, the d-band center of the Pd/NiMoO₄/NF drops to −5.25 eV compared with that of Pd/NF (−4.87 eV), although the d-band center of the NiMoO₄/NF (−5.75 eV) is the lowest, the catalyst has no obvious EG oxidation activity at low potential, so it will not be discussed. Furthermore, as shown in Supplementary Fig. 12, the d-band center of Pd/NF and Pd/NiMoO₄/NF are calculated to be −1.83 eV and −2.45 eV, respectively, which agree well the results of the surface valance band photoemission spectroscopy. The downshift of d-band center is contributed to the desorption of carbonyl intermediates (such as glycolate acid) to achieve long-term stability and high selectivity of the target products[28].

## Electrocatalytic performances for ethylene glycol oxidation

To investigate the electrocatalytic performance of as-prepared Pd/NiMoO₄/NF as well as control samples for ethylene glycol oxidation at the anode, a series of electrochemical measurements were performed in a three-electrode setup. Supplementary Fig. 13 presents the linear sweep voltammetry (LSV) curves of Pd/NiMoO₄/NF in 1.0 M NaOH with or without 1 M ethylene glycol. In the absence of ethylene

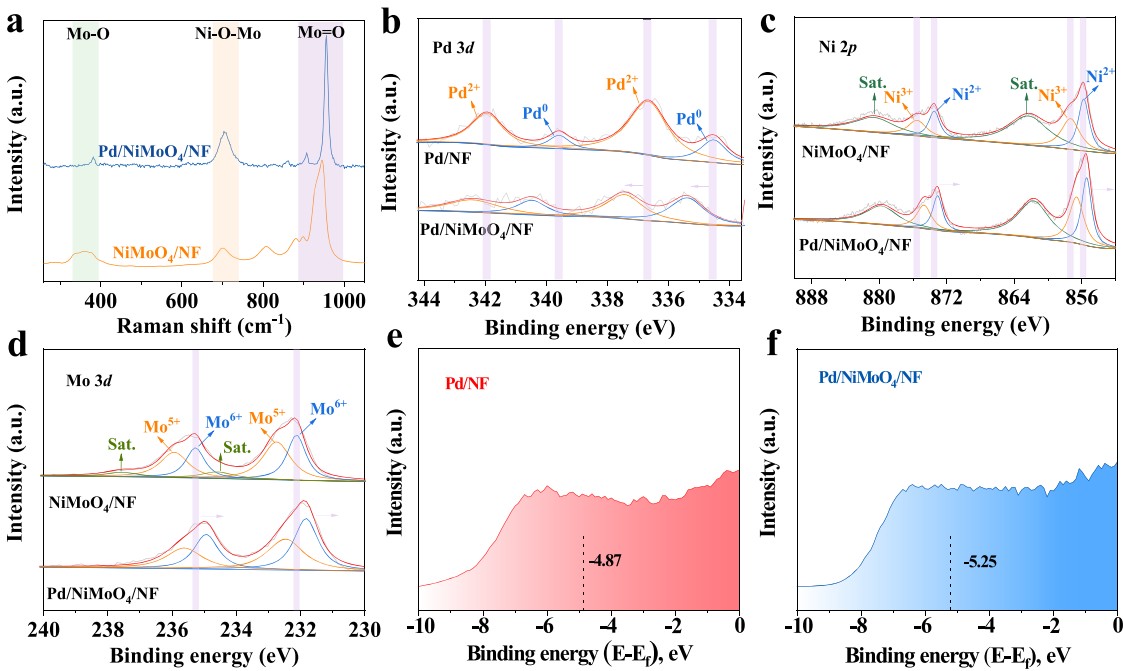

**Fig. 3 | Structure and electronic properties of catalysts. a** Raman spectras of Pd/NiMoO₄/NF and NiMoO₄/NF. **b** High-resolution XPS spectra of Pd 3d (**c**) Ni 2p and (**d**) Mo 3d of Pd/NiMoO₄/NF and NiMoO₄/NF. **e** XPS valence band spectra of Pd/NiMoO₄/NF and (**f**) Pd/NF.

glycol, the electrode shows a moderate OER activity, which reaches the anodic current density of 100 mA cm$^{-2}$ at the potential of 1.61 V versus reversible hydrogen electrode (vs. RHE). When introducing 1 M ethylene glycol, the current density increases markedly, and the anodic potential strikingly decreases to 0.79 V vs. RHE at 100 mA cm$^{-2}$, which is 820 mV lower than that of OER. Also, to give a more detailed comparison with OER, Supplementary Fig. 14 displays that the anodic potentials in ethylene glycol solution are reduced by at least 820 mV at the current density of 10, 50, 100, 200, and 300 mA cm$^{-2}$. Noticeably, Pd/NiMoO$_4$/NF shows significantly higher performance than those of the Pd/NF, NiMoO$_4$/NF, and pure NF references for electrocatalytic ethylene glycol oxidation (Fig. 4a), which indicates that the interaction between Pd and NiMoO$_4$ plays a crucial role in enhancing the EGOR activity. The effects of the concentration of NaOH and ethylene glycol in the electrolyte on the oxidation performance of ethylene glycol at the anode were also studied, as shown in Supplementary Fig. 15. The highest EG oxidation activity (1.2 A cm$^{-2}$) is achieved in 9 M NaOH solution containing 1 M EG. Nevertheless, in order to avoid excessive waste of NaOH, we

choose 1 M NaOH solution added with 1 M EG. What's more, the performance of catalysts with different Pd content and oxide substrates is further discussed (Supplementary Figs. 16–19). As shown in Fig. 4b, Pd/NiMoO$_4$/NF shows much lowered Tafel slop (207.7 mV dec$^{-1}$) compared to that of Pd/NF (219.4 mV dec$^{-1}$), indicating the faster reaction kinetics of Pd/NiMoO$_4$/NF at the anode. Besides, electrochemical impedance spectra (EIS) of Pd/NiMoO$_4$/NF, Pd/NF, NiMoO$_4$/NF and NF at the open-circuit voltage were obtained. Pd/NiMoO$_4$/NF delivers a smaller charge transfer resistance (Rct) than Pd/NF, NiMoO$_4$/NF and pure NF, (Supplementary Fig. 20), which implies the rapid charge transfer kinetics of Pd/NiMoO$_4$/NF.

To better understand the origin of the markedly high EGOR performance of Pd/NiMoO$_4$/NF, the electrochemical double-layer capacitance ($C_{dl}$) has been obtained to calculate the electrochemically active surface area (ECSA) (Fig. 4e and Supplementary Fig. 21). Pd/NiMoO$_4$/NF shows a similar $C_{dl}$ value (7.32 mF cm$^{-2}$) with Pd/NF (9.52 mF cm$^{-2}$), which is apparently higher than those of NiMoO$_4$/NF (2.33 mF cm$^{-2}$) and NF (0.83 mF cm$^{-2}$). The ECSA-normalized EGOR activities have also been obtained, which are in the order of

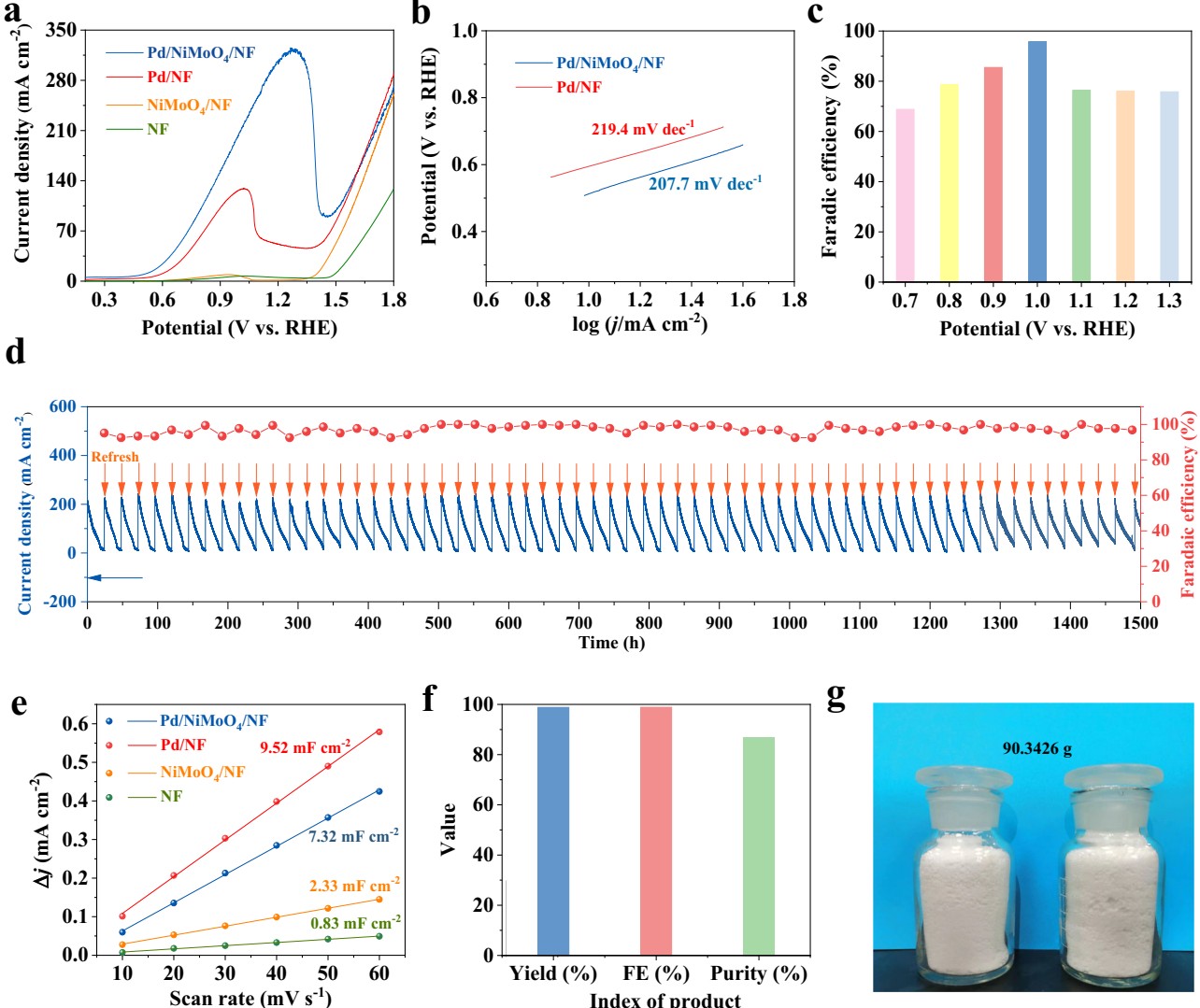

**Fig. 4 | Electrocatalytic performances of Pd/NiMoO$_4$/NF toward ethylene glycol oxidation at the anode. a** LSV curves of obtained electrocatalysts for ethylene glycol anodic oxidation. **b** Tafel slopes of Pd/NiMoO$_4$/NF and Pd/NF. **c** Faradaic efficiencies (FEs) of Pd/NiMoO$_4$/NF for sodium glycolate production for 2 h chronoamperometry at varied potentials. **d** FEs and current density of Pd/NiMoO$_4$/ NF for sodium glycolate production for 1500 h electrolysis cycles. **e** Double layer capacitance (Cdl) of Pd/NiMoO$_4$/NF, NiMoO$_4$/NF, Pd/NF, and pure NF. **f** The yield, FE and purity of GA obtained from EGOR. **g** The optical image of obtained GA product.

Pd/NiMoO$_4$/NF > Pd/NF > NiMoO$_4$/NF > NF (Supplementary Fig. 22), further demonstrating largely elevated EGOR activity of the Pd/NiMoO$_4$/NF catalyst.

While achieving high EGOR activity is important, good stability may play an even more important role in EGOR for further large-scale applications. Subsequently, a long-term chronoamperometry test using Pd/NiMoO$_4$/NF as the electrocatalyst was carried out at 0.7–1.3 V vs. RHE to determine the obtained ethylene glycol oxidation products at the anode by $^1$H NMR and $^{13}$C NMR spectroscopy (Supplementary Figs. 23, 24). As shown in the $^1$H NMR spectra (Supplementary Fig. 23), sodium glycolate (3.89 ppm) is the primary EG oxidation product. Besides, only small amounts of formate (FA) were detected. The highest faradaic efficiency (FE) of 95.7% was obtained at 1.0 V vs RHE, therefore this voltage was chosen as the optimum potential (Fig. 4c and Supplementary Fig. 25). With The FEs varies in the range of 93.1% ‑ 99.9% for glycolic acid production and the total charges have been maintained for 1500 h of intermittent chronoamperometry (see details in Methods) (Fig. 4d and Supplementary Fig. 26)[36]. No significant decreases in FE and consumed charges can be observed during the 1500 h stability tests, elucidating the robust durability of the Pd/NiMoO$_4$/NF electrocatalyst. As far as we know, this is the longest stability test among literature reports. (Supplementary Fig. 27)[20–22,24,27,32,36–44]. In contrast, during cyclic electrolysis, Pd/NF catalyst shows a much lower FE toward glycolic acid (87-88%), accompanied by quick decays of the current density, indicating the critical role of NiMoO$_4$ in enhancing the selectivity and stability towards EGOR (Supplementary Fig. 28).

The development of economic separation approaches is essential to obtaining the value-added products and the industrial applications of electrocatalytic oxidations. The previously reported separation methods necessitate a large amount of strong acids to neutralize excess strong bases retained in the electrolyte, which inevitably leads to the wastes of both acids and bases. Here, based on the stoichiometric EG and NaOH components in the electrolyte before reaction and in-situ acid-base reaction, the pH of the electrolyte after a cycle test reaches about 10.08 (Supplementary Fig. 29) due to the hydrolysis of sodium glycolate, which implies negligible alkaline retained in the electrolyte. Therefore, no additional acid is needed for the product separation, which enables full use of electrolyte and avoids the waste of acid and base. After 1500 hours of cyclic electrolysis, 90.34 g sodium glycolate was successfully isolated at a purity of 86.75% and a yield of 98.91% (Fig. 4f, g and Supplementary Fig. 30). Besides, the Pd/NiMoO$_4$/NF electrode after ethylene glycol electro-oxidation for 1500 h was further characterized by XRD, SEM, EDS, TEM and HRTEM to evaluate the stability of the catalyst. No significant changes in the XRD pattern of the catalyst can be found (Supplementary Fig. 31). Moreover, the SEM image of Pd/NiMoO$_4$/NF shows maintained original morphology after ethylene glycol oxidation stability tests (Supplementary Fig. 32), highlighting the superior structural robustness. SEM-EDS elemental mapping analysis displays that the Pd, Ni, Mo, and O are homogeneously distributed throughout the Pd/NiMoO$_4$/NF (Supplementary Fig. 33). In addition to this, both the Pd and NiMoO$_4$ lattice can be found in the HRTEM image (Supplementary Fig. 34). These results all suggest the excellent stability of the obtained catalyst. Furthermore, the produced H$_2$ amount was quantified by gas chromatography (GC). Supplementary Fig. 35 shows the standard curve of H$_2$ production measured by GC. As exhibited in Supplementary Fig. 36, the Faradaic efficiency for H$_2$ at varied consumed charges have been calculated to be close to 100% and the purity of H$_2$ is 99.9%.

## Understanding the mechanism

The adsorption of EG species is important for the electrocatalytic EG oxidation in alkaline electrolyte. Figure 5a and Supplementary Fig. 37 shows the open circuit potential (OCP) measurement results, which can be applied to show the influences of organic absorbates on the

inner Helmholtz layer[15]. Upon adding 1 M EG, the OCP is significantly decreased for Pd/NiMoO$_4$/NF ($\Delta$ = 0.68 V) compared with Pd/NF ($\Delta$ = 0.45 V) and NiMoO$_4$/NF ($\Delta$ = 0.09 V), indicating that EG is more easily adsorbed on the Pd/NiMoO$_4$/NF than on Pd/NF and NiMoO$_4$/NF surface. At the same time, the weak adsorption of glycolate is the key to avoid over-oxidation. As shown in Fig. 5b, upon the addition of 1 M GA, the OCP changes is negligible ($\Delta$ = 0.03 V) compared with that of 1 M EG ($\Delta$ = 0.68 V), which indicates that GA is weakly adsorbed in the inner Helmholtz layer so as to prevent its over-oxidation. Furthermore, the in-situ EIS measurements were applied to explore the catalytic kinetics and the electrode/electrolyte interface properties at varied potentials[45,46]. The Nyquist and Bode plots of Pd/NiMoO$_4$/NF in 1 M NaOH containing 1 M EG are shown in Fig. 5c, d. A phase peak shift at 0.45 V (vs. RHE) in the low-frequency can be clearly observed, indicating the beginning of the reaction at this potential. As the potential increases, the phase angle peaks become weakened and shift towards higher frequency, demonstrating that more and more adsorbed EG are rapidly oxidized at a fast interfacial charge transferring rate. As shown in Supplementary Fig. 38, the oxidation of GA shows an opposite trend to that of EG, which also implies a weaker adsorption of GA on the Pd/NiMoO$_4$/NF surface. The above results demonstrate a poor catalytic activity of Pd/NiMoO$_4$/NF toward GA oxidation reaction. Besides, to further verify the above results, the oxidation activity of GA in 1 M NaOH has also been explored, which is even negligible compared with that of EG in the same solution (Supplementary Fig. 39). The markedly weaker adsorption of GA than that of EG on Pd/NiMoO$_4$/NF will ensure the over-oxidation prevention and selectivity enhancement of GA in comparison with EG.

Similarly, the adsorption of OH$^-$ species is also crucial for the electrocatalytic EG oxidation in alkaline electrolyte. Figure 5e shows the representative OH adsorption and desorption curves on Pd in Pd/NiMoO$_4$/NF and Pd/NF. As shown in Fig. 5e, Pd/NiMoO$_4$/NF exhibits OH$^-$ adsorption/desorption bands at significantly lower onset potential (0.5 V vs RHE) than Pd/NF (0.6 V vs RHE), which is attributed to the introduced oxyphilic Ni species in NiMoO$_4$, which can efficiently oxidize OH$^-$ to *OH adsorbed on Pd surfaces at a relatively low potential. At the same time, combined with the XPS results, the electron-rich substrate NiMoO$_4$ can absorb no OH$^-$ anions on its surface, while the positively charged Pd nanosheets can capture OH$^-$ to facilitate the further oxidization of *C. The important roles of *OH$_{ad}$ species were further explored by CO-stripping Cyclic Voltammetry (CV) measurements[47]. A distinct CO oxidation peak appears for Pd/NF and NiMoO$_4$/NF in the first anodic scan at the peak potentials at 1.090 V vs. RHE and 0.785 V vs. RHE (Supplementary Figs. 40, 41). After introducing NiMoO$_4$, the peak potentials of CO oxidation decrease to a lowered potential 1.026 V vs RHE (Fig. 5f) for Pd/NiMoO$_4$/NF, which indicates that CO is weakly adsorbed on the surface and can be easily oxidative-removed on account of the downshift of the d-band center of Pd, eventually contributing to the observed superior EGOR stability.

Moreover, in-situ electrochemical Fourier Transform Infrared (FTIR) spectroscopy has been employed to further understand the origin of the high EGOR performance of Pd/NiMoO$_4$/NF. The electrocatalysts were placed on the flat plane of a Si hemicylindrical prism and pressed by a glass carbon electrode. The FTIR spectra were recorded in 1 M NaOH aqueous solution with 1 M EG at the potential of −0.9 to 0.1 V vs Hg/HgO. As displayed in Supplementary Fig. 42, a peak at 1868 cm$^{-1}$ corresponding to the multiple bonded CO(CO$_M$) can be well observed on Pd/NF; however, CO$_M$ signal can be hardly detected for Pd/NiMoO$_4$/NF over the whole potential range, further demonstrating high CO tolerance of Pd/NiMoO$_4$/NF, which is consistent with the CO stripping experimental results. Apparently, a downward enhancement band at 1076 cm$^{-1}$ can be observed, which is attributed to the stretching vibration of aldehyde, indicating that EG is firstly oxidized to glycolaldehyde species[20]. Further, the peaks at 1326 cm$^{-1}$ and 1581 cm$^{-1}$, belonging to symmetric and antisymmetric stretching bands of

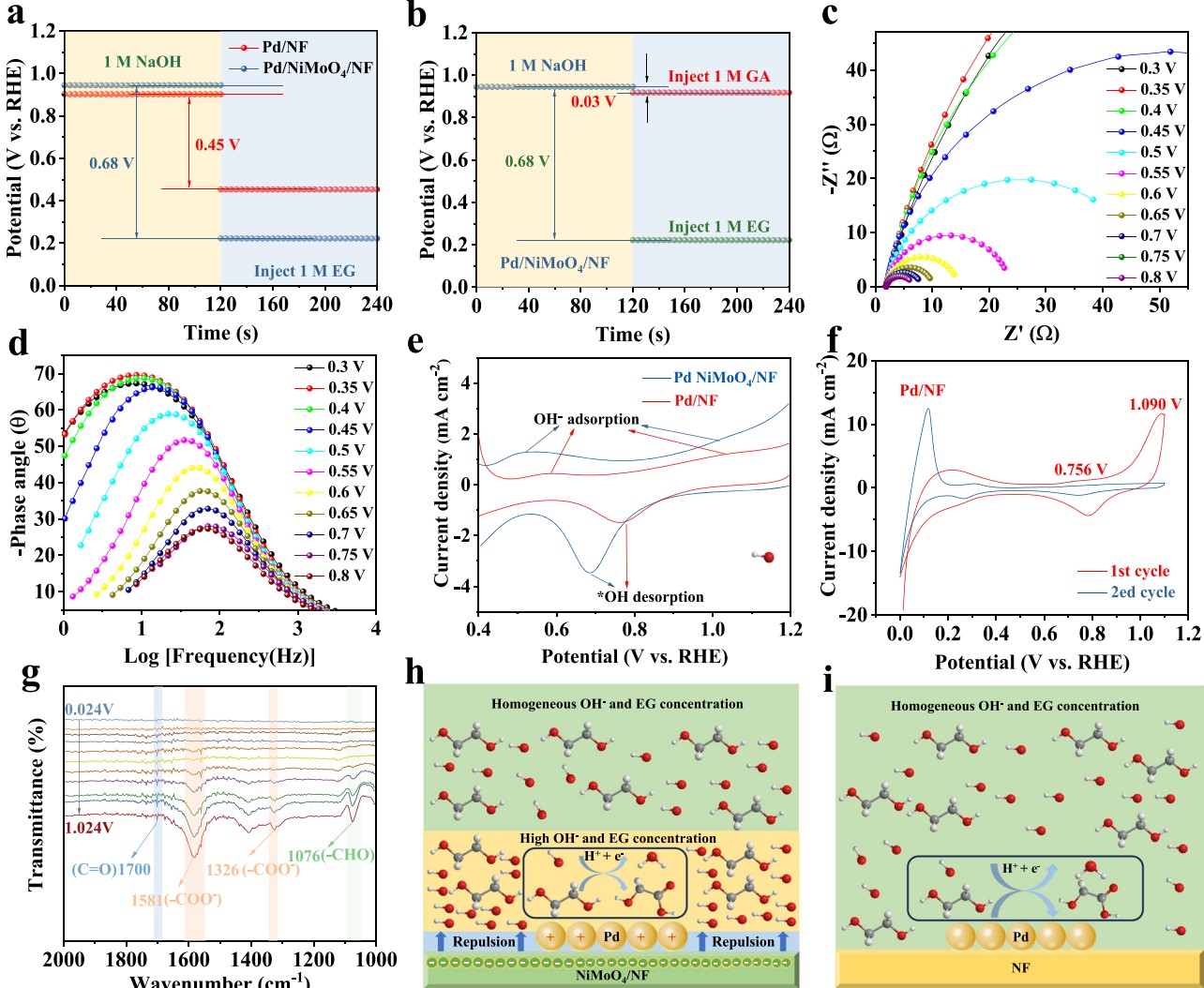

**Fig. 5 | Understanding the mechanism. a** Open circuit potentials (OCPs) of Pd/NiMoO$_4$/NF and Pd/NF in 1 M NaOH solution before and after EG was added. **b** OCPs of Pd/NiMoO$_4$/NF in 1 M NaOH solution with EG or GA addition. **c** The Nyquist plots and (**d**) corresponding Bode phase plots of Pd/NiMoO$_4$/NF electrode at varied potentials in 1 M NaOH with 1 M EG. **e** CV curves of Pd/NiMoO$_4$/NF and Pd/NF in 1 M NaOH. **f** CO stripping experiments of Pd/NiMoO$_4$/NF. **g** In-situ electrochemical FTIR spectra of EGOR catalyzed by Pd/NiMoO$_4$/NF. **h** Schematic illustration of the proposed EGOR mechanism on Pd/NiMoO$_4$/NF and (**i**) Pd/NF.

carboxyl group of GA, respectively, indicates the further oxidation of glycolaldehyde to GA[21]. Besides, the distinct vibration peaks at 1700 cm$^{-1}$ and 1717 cm$^{-1}$ have been detected on Pd/NiMoO$_4$/NF and Pd/NF, respectively, which can be assigned to the stretching vibration of carbonyl group (C = O), indicating the formation of 2-hydroxyacetyl (*OC-CH$_2$OH) intermediates (Fig. 5g).

Furthermore, the Density functional theory (DFT) calculation is also implied to investigate the reaction mechanism. The optimized models of EG oxidation on the Pd/NiMoO$_4$ and Pd catalysts are shown in Supplementary Figs. 43, 44. Besides, the corresponding Gibbs free energy profile at 0 V vs RHE were calculated. As shown in Supplementary Fig. 45, obviously, we find that the addition of *OH to form the co-adsorption of *OH and *OCH$_2$CH$_2$OH is the rate-determining step (RDS) in the whole reaction from EG to the glycolic acid on Pd/NiMoO$_4$ and Pd catalysts. In particular, the free-energy change of the RDS on Pd/NiMoO$_4$ (0.58 eV) is much lower than 0.86 eV of Pd, suggesting that the Pd/NiMoO$_4$ catalysts could adsorb OH at a relatively lower onset potential, which is in accordance with the results of CV.

What's more, As shown in Supplementary Fig. 10, the charge density difference results indicate that 1.63 electrons transfer from the Pd to the NiMoO$_4$ substrate, which suggests that the Pd is positively charged and the NiMoO$_4$ substrate is negatively charged. The positively charged Pd could attract anions OH$^-$ to promote the hydrogen abstraction of EG. The calculated adsorption energies of OH on the surfaces of Pd/NiMoO$_4$ are 0.20, 0.26, and 0.28 eV at Pd, Ni, and Mo adsorption sites, respectively (Supplementary Fig. 46). The above results indicate that the addition of OH is at the Pd site. Besides, as shown in Supplementary Fig. 47, the calculated adsorption energies of EG on Pd (111) and Pd/NiMoO$_4$ were −0.57 and −0.98 eV, respectively, which indicates that the EG is more easily adsorbed on the Pd/NiMoO$_4$, in accordance with the results of OCP measurements. Besides, the calculated adsorption energies of EG on the surfaces of Pd/NiMoO$_4$ are −0.98, −0.53, and −0.36 eV at Pd, Ni and Mo adsorption sites, respectively (Supplementary Fig. 48), indicating that the EG is more favorable for adsorption at Pd site. Furthermore, as shown in Supplementary Fig. 28, the main products of EGOR catalyzed by Pd/NF is sodium glycolate; however, the NiMoO$_4$/NF has no obvious EG oxidation activity at low potential. The above results indicate that the abstraction of hydrogen is also at the Pd site.

According to above results, the reaction mechanism and path of EG has been proposed as following (Fig. 5h and Supplementary Fig. 49). The surface of NiMoO$_4$ becomes negatively charged due to

the transferring of electrons from Pd to $NiMoO_4$, which can promote migration of hydroxyls away from the $NiMoO_4$ surface and result in the enhanced local hydroxyl concentration around the Pd surface. In contrast, the hydroxyls on the Pd surface are uniformly distributed on Pd/NF catalysts (Fig. 5i). The higher local hydroxyl concentration around the Pd surface can promote the faster conversion of ethylene glycol and activation of CO, thus favoring the enhancement of EGOR activity and stability. In addition, the introduction of $NiMoO_4$ can also promote the adsorption of EG, and the higher local concentration facilitates its faster conversion.

### Membrane-free flow electrolyzer performance

To demonstrate its economic feasibility in a two-electrode system, a homemade membrane-free flow electrolyzer has been assembled by using $Pd/NiMoO_4/NF$ catalyst as the anode and pre-treated nickel foam as the cathode at a working area of 1 $cm^2$ (Fig. 6a). A homogenous electrolyte (1 M NaOH containing 1 M ethylene glycol) was circulated in the flow reactor by using a peristaltic pump at a flow rate of 40 mL $min^{-1}$. The LSV curves (Fig. 6b) shows that electro-oxidation of ethylene glycol occurs from ~0.5 V and reaches the current density of 115 mA $cm^{-2}$ at 1.2 V in flow cell, which is higher than that in the corresponding three-electrode cell, demonstrating priomising application potentials of ethylene glycol electrooxidation in flow cell.

We then performed intermittent potential tests to evaluate the performance of sodium glycolate production in the electrolyzer[36]. After 23 hours of stability test (Fig. 6c), the conversion of ethylene glycol reaches 50%, and the FE and selectivity of sodium glycolate are 93.1% and 93.6%, respectively. Finally, the sodium glycolate crystal was obtained as confirmed by XRD (Fig. 6d).

## Conclusions

In summary, an electrocatalyst $Pd/NiMoO_4/NF$ with rather low Pd loading (3.5 wt.%) has been obtained for efficient electrocatalytic ethylene glycol oxidation, which features the faradaic efficiency, yield of sodium glycolate and stability of as high as 98.9%, 98.8%, and 1500 h, respectively. More importantly, the obtained glycolic acid has been converted to value-added sodium glycolate by in-situ acid-base reaction with NaOH electrolyte, which favors the enhancement of

atom utilization and the separation of sodium glycolate separation. The weak adsorption of sodium glycolate on the catalyst surface plays a significant role in avoiding excessive oxidation and achieving high selectivity. The electron transfer between Pd and $NiMoO_4$ and the downshift of the d band-center of Pd have been proposed to promote the adsorption of OH* and the oxidation removal of CO, thereby leading to the enhanced activity and stability of $Pd/NiMoO_4/NF$.

## Methods

### Chemicals

Nickel nitrate hexahydrate ($Ni(NO_3)_2 \cdot 6H_2O$, 99%), Sodium molybdate dihydrate ($Na_2MoO_4 \cdot 2H_2O$, 99.99%), and Sodium borohydride ($NaBH_4$, 99.99%) were purchased from China National Pharmaceutical Group Co., Ltd. Sodium tetrachloropalladate ($Na_2PdCl_4$) was purchased from Shanghai Titan Technology Co. Ltd. Sodium hydroxide (NaOH, 99.0%) were from Macklin. Nickel Foam (NF, 0.15 mm thick) was purchased from Tianjin Gaoshi Ruilian Photoelectric Technology Co., Ltd. Maleic acid (99%, RG, grade) and deuterated water ($D_2O$) were bought from Adamas Reagent Co., Ltd. Ethylene glycol (AR) was bought from Chinese medicine reagent.

All chemicals were used as received without any further purification. Deionized water (DIW) was used in all experiments.

### Synthesis of the $NiMoO_4$ nanorods/NF

$NiMoO_4/NF$ was prepared as a precursor using hydrothermal synthesis. First, NF (3 cm*3 cm) was ultrasonically cleaned in 3 M HCl, ethanol, and deionized water to remove any surface impurities. Thereafter, Ni $(NO_3)_2 \cdot 6H_2O$ (1 mmol) and $Na_2MoO_4 \cdot 2H_2O$ (1 mmol) were dissolved in deionized water (30 mL) and stirred for 10 min. Subsequently, the as-obtained precursor solution and a piece of NF were placed in a Teflon-lined stainless-steel autoclave and maintained at 160 °C for 6 h. After the autoclave was cooled to room temperature, the NF with light green precipitates on the surface was taken out and washed with deionized water and ethanol respectively to remove any unreacted residues before being fully dried at 60 °C overnight under vacuum. In order to obtain crystallized $NiMoO_4$ nanostructures, the conductive substrates with as-grown precursor hierarchical structures were calcined at 450 °C for 2 h with a temperature ramp rate of 1 °C $min^{-1}$ in an argon atmosphere.

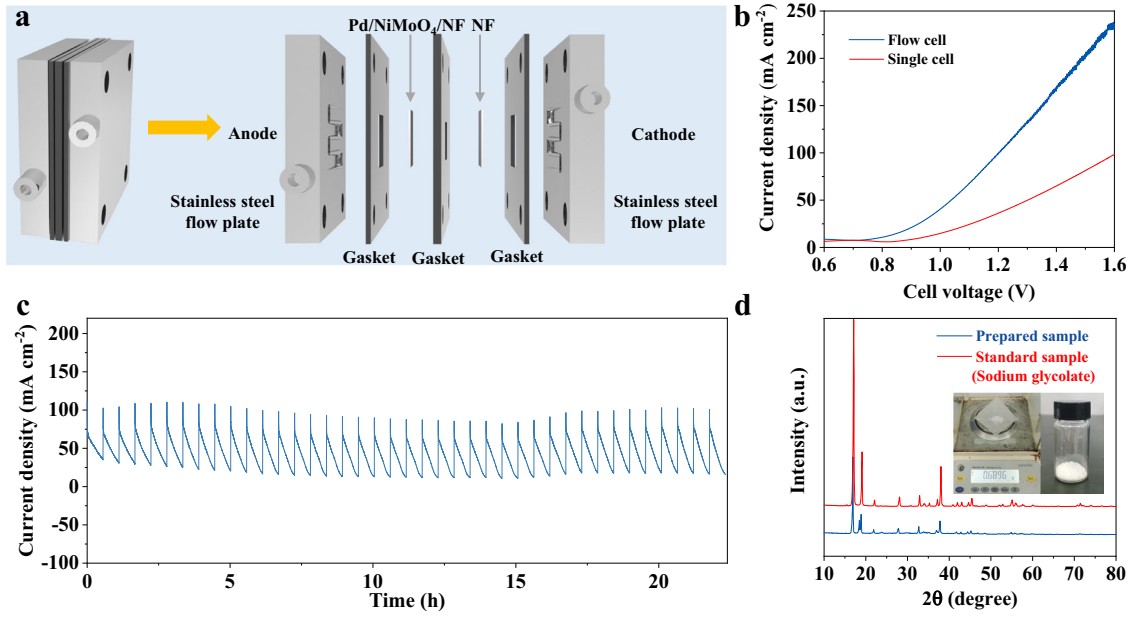

**Fig. 6 | Membrane-free flow cell for EG oxidation. a** The MEA setup for paired HER (−) // EG oxidation (+). **b** LSV curves of $Pd/NiMoO_4/NF$ in 1 M NaOH with 1 M EG in single cell or flow cell. **c** Stability tests of $Pd/NiMoO_4/NF$ toward EGOR. **d** XRD pattern of self-prepared sodium glycolate.

### Synthesis of the Pd/NiMoO₄/NF

The NiMoO₄/NF (2 cm*3 cm) and 20 mg Na₂PdCl₄ were dispersed in 30 mL of water and stirred for 12 h. The black sample was generated after adding 1 mL of NaBH₄ (0.15 M) and stirring for 4 h. Subsequently, the resulting sample was washed via deionized water and ethanol and further dried at 60 °C for 10 h in a vacuum to obtain Pd/NiMoO₄/NF catalysts. Pd/NF was synthesized in a similar manner using the pre-treated NF (2 cm*3 cm) as the precursor.

### Materials characterization

X-ray diffraction (XRD) patterns were recorded on a Rigaku D/MAX 2550 diffractometer at 35 kV and 25 mA using Cu Kα radiation (I = 1.5418 Å). Scanning electron microscope (SEM) images and energy-dispersive elemental mapping (EDS) images were acquired using a ZEISS Gemini 450. Transmission electron microscopy (TEM) images were acquired on a JEM-2100F at 200 kV. X-ray photoelectron spectroscopy (XPS) was performed on AXIS SUPRA with an Mg Kα radiation source (hν = 1253.6 eV). The position of the C 1s peak at 284.8 eV was utilized as a calibration reference for determining the precise binding energy (±0.1 eV). Inductively coupled plasma emission spectroscopic (ICP-OES) analysis was recorded on Agilent 700 Series instrument. Electrochemical in situ FTIR measurements were performed on a Linglu instruments ECIR-II cell mounted on a Pike Veemax III ATR using a single crystal bouncing silicon crystal. The binding energy values were referenced using the C 1s peak position at 284.8 eV. The d-band centers were evaluated by applying the following formula.

$$d - \text{band center} = \int_{-10eV}^{oeV(E_f)} (\text{binding energy(E)}$$
$$\times \text{intensity(E)})dE / \int_{-10eV}^{oeV(E_f)} \text{intensity(E)}dE$$

### Electrochemical measurements

All measurements for OER and ethylene glycol oxidation were conducted on a BioLogic VSP-300 electrochemical workstation in a three-electrode single cell at room temperature with 1*1 cm² of the as-made materials as the working electrode, a Pt plate as the counter electrode, and HgO/Hg (1.0 M NaOH) as the reference electrode, respectively. The HgO/Hg (1 M NaOH) reference electrode was calibrated with respect to RHE in high-purity hydrogen-saturated 1 M NaOH electrolytes, two platinum wire electrodes were used as the counter electrode and the working electrode in this measurement, respectively. The conversion formula of the Hg/HgO reference electrode is: E (RHE) = E (Hg/HgO) + 0.059*pH+0.098. Polarization curves were obtained using LSV with a scanning rate of 10 mV s⁻¹ in the region of −0.8–0.9 V (vs HgO/Hg). Cyclic Voltammetry (CV) was measured using CV scans from −0.1 V to 1.2 V vs. RHE at a scan rate of 10 mV s⁻¹. The durability test was performed by chronoamperometry at 1.0 V vs. RHE. EIS measurements were performed in the frequency range of 100 kHz to 10 MHz with an alternating current voltage of 5 mV. The electrochemical double layer capacitances (C_dl) of various samples were confirmed by CV in the potential region without faradaic process to calculate the ECSA. EGOR tests were conducted in a 1 M NaOH solution containing 1 M ethylene glycol. The area of the working electrode in the electrolyte was fixed at 1*0.7 cm² and all current densities were normalized to the geometrical area of the electrode. All the curves were used without IR compensation.

### CO stripping voltammetry experiments

The CO stripping curves were carried out in a 1 M NaOH solution. Prior to the tests, the 1 M NaOH solution was first deaerated with high-purity N₂ (99.999%) for 30 min. Then, pure CO gas was bubbled into the solution for 30 min while the potential of the working electrode was held at a constant potential of 0.1 V vs. RHE to ensure effective CO

adsorption on the electrode surface. Nonadsorbed CO molecules were then repelled by bubbling N₂ gas in the solution for 20 min. Subsequently, CO stripping curves were initiated from ∼0.0 V to 1.1 V vs. RHE in the anodic direction at a scan rate of 50 mV s⁻¹ for at least two consecutive cycles.

### Product quantification

The chronoamperometry tests were conducted at 1.0 V vs. RHE (for the three-electrode system) or at the cell voltage of 1.2 V (for the two-electrode system). After 2 h of electrolysis, the products were analyzed and quantified by ¹H and ¹³C nuclear magnetic resonance (¹H and ¹³C NMR). ¹H and ¹³C NMR spectra were recorded on an Avance II 300 instrument (Bruker). 500 μL electrolyte was collected and diluted with 100 μL of D₂O. Maleic acid was used as an internal standard. A Ramin GC2060 gas chromatograph with a packed column and a thermal conductivity detector was used to quantify the generated H₂ generated at the cathode during electrolysis. The standard curve of H₂ was exhibited in Supplementary Fig. 31. The amount of theoretically generated H₂ was calculated as $Q_{tot} \times V_m / (Z \times F)$ ($Q_{tot}$ is the total charge passed through the electrodes, $V_m$ is the molar volume of gas, Z = 2 is the number of electrons needed to produce a molecule of H₂, F = 96485 C mol⁻¹ is the Faraday constant). The formula used to calculate the purity of sodium glycolate is:

$$\text{Purity} = m1/m2$$

m1 refers to the mass of sodium glycolate is actually separated, m2 refers to the total mass of the isolated product.

The yield (%) and selectivity (%) of glycolic acid formation can be determined by the following Eqs. (1) and (2), respectively

$$\text{Yield(\%)} = \{N(\text{glycolic acid yield})/4*N(\text{initial ethylene glycol})\}*100\% \quad (1)$$

$$\text{Selectivity(\%)} = \{N(\text{glycolic acid yield})/4*N(\text{ethylene glycol consumed})\}*100\% \quad (2)$$

The Faraday efficiency (%) of the glycolic acid and H₂ production can be determined by the following Eqs. (3) and (4), respectively

$$\text{FE(\%)} = \{N(\text{glycolic acid yield})/\{Q_{totl1}/(Z_1*F)\}\}*100\% \quad (3)$$

$$\text{FE(\%)} = \{N(H_2 \text{ Production})/\{Q_{totl2}/(Z_2*F)\}\}*100\% \quad (4)$$

Where $Q_{totl1}$ and $Q_{totl2}$ are the total charge passed through the electrodes, $Z_1 = 4$ is the number of electrons that form a mole of glycolic acid, $Z_2 = 2$ is the number of electrons that produce a molecule of H₂, F is the Faraday constant (96,485 C mol⁻¹).

## Data availability

All data is available from the authors upon request.

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

## Acknowledgements

This work was supported by National Key R&D Program of China (2022YFB4002700), Shanghai Science and Technology Committee Rising-Star Program (22QA1403400) and the Natural Science Foundation of Shanghai (21ZR1418700). The authors would like to thank ECNU Multifunctional Platform for Innovation for support of TEM character-izations (004). The authors extend their gratitude to Shiyanjia Lab (www.shiyanjia.com) for providing in valuable assistance with the SEM analysis.

## Author contributions

J.S. and L.C. led the project. K.S. designed and performed the experiments; X.T. and D.S. contributed considerably to the revision of the manuscript. All authors discussed the results and commented on the manuscript.

## Competing interests

The authors declare no competing interests.
