## [Peer Review File · Nature Communications]

Pd/NiMoO₄/NF electrocatalysts for the efficient and ultra-stable synthesis and electrolyte-assisted extraction of glycolateREVIEWER COMMENTS

Reviewer #1 (Remarks to the Author):

In this work, the authors report the electrocatalytic conversion of ethylene glycol to sodium glycolate via an in-situ acid-base reaction with ultra-stable (1500 h) electrocatalytic stability, and significantly high yield (98.8%) and Faradaic efficiency (98.9%) of sodium glycolate, which is an interesting and impressive research work. In addition, the TEA results show that when the current density is 100 mA/cm², the Faradic efficiency of sodium glycolate is more than 80%, this process can be profitable, which is promising for further practical applications. Besides that, the research includes electrochemical in-situ FTIR, and in-situ EIS for detailed mechanism investigation. Therefore, I would like to recommend its publication in Nature Communication after addressing the following comments.

1. The last sentence of the abstract only reflects the experimental results of this study. Authors should emphasize the profound significance of this research.
2. The author mentioned the purity of sodium glycolate in the article, please provide the calculation formula of sodium glycolate purity.
3. The author only provided the ¹H NMR spectra of sodium glycolate in the article, please provide the ¹³C NMR spectra of the product.
4. Please provide the conversion formula of the Hg/HgO reference electrode used by the author in the article.
5. Although the authors cited sufficient recent works in this field, some updated reports should be cited.

Reviewer #2 (Remarks to the Author):

In this article, the authors reported an efficient catalyst Pd/NiMoO₄/NF for the oxidation of ethylene glycol to glycolic acid. The reported faradaic efficiency and the yield of sodium glycolate reached 98.9 % and 98.8 %. The authors proposed that the electron transfer between Pd and NiMoO₄ and the decrease in the d-band center of Pd promoted the adsorption of OH* and the oxidation removal of CO, resulting in an increase in the activity and stability of Pd/NiMoO₄/NF. This explanation is almost the same with existing articles (Angew. Chem. Int. Ed. 2023, 62, e202300094). While the performance is encouraging, I do not think the paper provide additional understanding for the field. So I suggest that the author switch to journals after examining the following issues.

1. The authors should supplement theoretical calculations to support the feasibility of experimental results, such as charge transfer, electronic structure, and reaction pathway.
2. The authors should compare the properties of three materials (Pd/NiMoO₄/NF, Pd/NF and NiMoO₄/NF), such as d-band centers, OCPs experiment, and CO stripping experiment etc.

Reviewer #3 (Remarks to the Author):

The manuscript submitted by Shi et al reports an interesting electrocatalysts Pd/NiMoO₄/NF for the production of sodium glycolate under ambient conditions. This electrocatalysts shows excellent selectivity and long-term stability. There are a few concerns this reviewer has for this work to be acceptable by Nature Communications, as detailed below.

1. In the introduction, the authors mentioned the importance of sodium glycolate, however no relevant literature was cited to support their claim. For instance, what is the global production of sodium glycolate per year and what is its market price per metric ton?
2. The authors indicated that a low Pd content (3.5%) was used to prepare this electrocatalyst, however it was not clear what this 3.5% is referring to, weight percentage or mole percentage? And what is the denominator in the calculation? The total mass of Pd/NiMoO₄/NF, or just Pd/NiMoO₄? It is probably to report the mass loading of Pd per geometric area of the electrode. In addition, why 3.5%? A detailed optimization of the Pd loading should be performed and reported.
3. The choice of NiMoO₄ as the support for the deposition of Pd is not well justified in the text. What is the rationale for using NiMoO₄, instead of NiO_x and MoO_x? Will the latter two materials show similar or inferior activity compared to NiMoO₄ once Pd is also loaded on them?
4. The detailed mechanism was not investigated in this work. The selectivity (not over oxidation) is critical for the success of this reaction. There are primarily two steps involved, hydrogen abstraction and hydroxide addition, during the alcohol transformation to carboxylate. It is not clear both steps take place on Pd, or NiMoO₄ is somehow also involved. Is the selectivity potential dependent? Based on Figure 4a, the maximum catalytic current appears beyond 1.2 V RHE, however the authors only conducted electrolysis till 1.1 V. Will increasing applied potential lead to more C-C bond cleaved products? Does the overall oxidation mechanism change when the applied potential is more positive?
5. Based on Figure 3e and 3f, it is not clear how the authors determine the d-band center. More explanations are needed.
6. Finally, the text should be thoroughly polished. There are quite a few typos or grammatical errors. For instance, line 18 on page 12, "...an electrocatalyst Pd/NiMoO₄/NF with lowered rather low Pd loading (3.5%)..." reads very odd.

Responses to the referees:

Reviewer #1: In this work, the authors report the electrocatalytic conversion of ethylene glycol to sodium glycolate via an in-situ acid-base reaction with ultra-stable (1500 h) electrocatalytic stability, and significantly high yield (98.8%) and Faradaic efficiency (98.9%) of sodium glycolate, which is an interesting and impressive research work. In addition, the TEA results show that when the current density is 100 mA/cm², the Faradic efficiency of sodium glycolate is more than 80%, this process can be profitable, which is promising for further practical applications. Besides that, the research includes electrochemical in-situ FTIR, and in-situ EIS for detailed mechanism investigation. Therefore, I would like to recommend its publication in Nature Communication after addressing the following comments. Some specific comments are listed as follows:

Response: Thank you very much for taking the time to review my manuscript and for your appreciation to this manuscript. All questions have been responded point by point as following, please review again.

1. The last sentence of the abstract only reflects the experimental results of this study. Authors should emphasize the profound significance of this research.

Response: Thank you very much for the kind suggestion. To emphasize the significance of this research, the last sentence of the abstract has been revised as following: "This work may pave a new way for the electrocatalytic conversions of biomass-derived alcohols through in-situ acid-base reaction as well as the design of efficient electrocatalysts." Related information has been added in the revised manuscript and marked yellow.

2. The author mentioned the purity of sodium glycolate in the article, please provide the calculation formula of sodium glycolate purity.

Response: Many thanks for the comment. The calculation formula of sodium glycolate purity is:

$$\text{Purity} = m1/m2$$

m1 refers to the mass of sodium glycolate calculated from NMR spectrum of the separated product, m2 refers to the total mass of the separated product. Related

information has been added in the revised manuscript and marked yellow.

3. The author only provided the ^1H NMR spectra of sodium glycolate in the article, please provide the ^{13}C NMR spectra of the product.

Response: Thank you very much for the kind suggestion. The ^{13}C NMR spectra of the product has been provided in the revised supporting information. As shown in **Figure S25**, the ^{13}C NMR spectrum of the electrolyte after 2 h anodic ethylene glycol oxidation on Pd/NiMoO₄/NF electrode shows obvious peaks at approximately 61.3 ppm and 180.2 ppm, which is identical with that of sodium glycolate, indicating the main oxidation product is sodium glycolate (62.7 ppm refer to the ethylene glycol). Related information has been added in the revised manuscript and marked yellow.

Figure S25. ^{13}C NMR spectra of the electrolyte before and after 2 h anodic ethylene glycol oxidation on Pd/NiMoO₄/NF electrode.

4. Please provide the conversion formula of the Hg/HgO reference electrode used by the author in the article.

Response: Thanks very much for the kind suggestion. The conversion formula of the Hg/HgO reference electrode is: $E(\text{RHE}) = E(\text{Hg}/\text{HgO}) + 0.059 \cdot \text{pH} + 0.098$, which has been added in the experiment part in the revised manuscript.

5. Although the authors cited sufficient recent works in this field, some updated reports should be cited.

Response: Thank you very much for the kind suggestion. In the revised version, updated references such as Ref. 18, Ref. 19 and Ref. 27 etc. have been added in the

revised manuscript and marked yellow.

Reviewer #2: In this article, the authors reported an efficient catalyst Pd/NiMoO₄/NF for the oxidation of ethylene glycol to glycolic acid. The reported faradaic efficiency and the yield of sodium glycolate reached 98.9 % and 98.8 %. The authors proposed that the electron transfer between Pd and NiMoO₄ and the decrease in the d-band center of Pd promoted the adsorption of OH* and the oxidation removal of CO, resulting in an increase in the activity and stability of Pd/NiMoO₄/NF. This explanation is almost the same with existing articles (Angew. Chem. Int. Ed. 2023, 62, e202300094). While the performance is encouraging, I do not think the paper provide additional understanding for the field. So I suggest that the author switch to journals after examining the following issues.

Response: Thank you very much for the kind suggestion. Besides the encouraging electrocatalytic performances toward ethylene glycol oxidation, we would like to address the highlights of this work which is totally different with reported references:

(1) In terms of performance: In this study, a Pd/NiMoO₄/NF electrocatalyst with much lowered Pd loading amount (3.5 wt.%) has been developed for the efficient, economic and ultra-stable glycolate synthesis. Pd/NiMoO₄/NF shows high Faradaic efficiency (98.9%), yield (98.8%), and ultrahigh stability (1500 h) towards electrocatalytic ethylene glycol oxidation, which is the longest durability test period ever-reported in the literatures so far.

(2) In terms of system design: The previously reported separation methods necessitate a large amount of strong acids to neutralize excess strong bases retained in the electrolyte, which inevitably leads to the wastes of both acids and bases. However, in our electrolysis process, the sodium hydroxide in the electrolyte can in-situ react with glycolic acid produced by the EG oxidation to produce sodium glycolate, which enables the feasible separation of such a target product without adding additional acid (the pH of the electrolyte after electrolysis is 10.08, indicating the full utilization of sodium hydroxide). Therefore, the electrolyte is fully utilized and the acid waste is minimized.

(3) In terms of reaction mechanism: Besides the decrease in the d-band center of Pd promoted the adsorption of OH* and the oxidation removal of CO, we further

employed OCP measurements, in-situ EIS and LSV to validate the weaker adsorption of sodium glycolate compared with EG on the catalyst surface, which avoids the excessive oxidation of sodium glycolate, and thus high product selectivity can be obtained.

Furthermore, following your suggestion, intensive experiments as well as DFT calculations have been conducted out and all questions have been responded point by point. We believe that by addressing the reviewer's suggestions and recommendations, the quality of our paper has been significantly improved. We hope it can meet the high standard of Nature Communications now.

1. The authors should supplement theoretical calculations to support the feasibility of experimental results, such as charge transfer, electronic structure, and reaction pathway.

Response: Thank you very much for the kind suggestion. Following your suggestion, we carried out theoretical calculations to support the experimental results.

(1) Charge transfer. As shown in **Figure S10**, the charge density difference results indicate that 1.63 electrons transfer from the Pd to the NiMoO₄ substrate. Moreover, as shown in the XPS results (**Figure 3b-d**), the peak of Pd 3d in Pd/NiMoO₄/NF shifts in a positive direction by 0.9 eV compared to that of the Pd/NF and the binding energies of Ni 2p and Mo 3d for Pd/NiMoO₄/NF are negatively shifted (~0.2 eV) compared with those of NiMoO₄/NF, which agree well the calculation results. The above results reveal the transferring of electrons from Pd to NiMoO₄, which further confirms the strong electronic interaction between Pd and NiMoO₄.

Figure S10. Charge density difference, where the blue and yellow isosurfaces denote the electron depletion and accumulation on Pd/NiMoO₄.

Figure 3. (b) High-resolution XPS spectra of Pd 3d (c) Ni 2p and (d) Mo 3d of Pd/NiMoO₄/NF and NiMoO₄/NF.

Electronic structure. As shown in **Figure S12**, the d-band center of Pd/NF and Pd/NiMoO₄/NF are calculated to be -1.83 eV and -2.45 eV, respectively, which agree well with the surface valance band photoemission spectroscopic results. The above results indicate the decrease in the d-band center of Pd.

Figure S12. PDOS (d-band) of Pd and Pd/NiMoO₄.

Reaction pathway. The optimized models of EG oxidation on the Pd/NiMoO₄ and Pd catalysts are shown in **Figure S43** and **S44**. Besides, the corresponding Gibbs free energy profile at 0 V vs RHE were calculated. As shown in **Figure S45**, obviously, we find that the addition of *OH to form the co-adsorption of *OH and *OCH₂CH₂OH is the rate-determining step (RDS) in the whole reaction from EG to the glycolic acid on Pd/NiMoO₄ and Pd catalysts. In particular, the free-energy change of the RDS on Pd/NiMoO₄ (0.86 eV) is much lower than 0.58 eV of Pd, suggesting that the Pd/NiMoO₄ catalysts can adsorb OH at a relatively lower onset potential, which is in accordance with the results of CV.

Figure S43. The optimized models of the EG oxidation on the Pd/NiMoO₄ catalysts.

Figure S44. The optimized models of the EG oxidation on the Pd catalysts.

Figure S45. Gibbs free energy landscapes for EGOR to glycolic acid over Pd/NiMoO₄ and Pd at 0 V vs RHE.

2. The authors should compare the properties of three materials (Pd/NiMoO₄/NF, Pd/NF and NiMoO₄/NF), such as d-band centers, OCPs experiment, and CO stripping experiment etc.

Response: Many thanks for the comment. Following your suggestion, the properties of three materials have been compared, including the d-band centers, OCPs experiments and CO stripping experiment etc.

(1) d-band centers. As shown in **Figure 3e-f** and **Figure S11**, the d-band center of the NiMoO₄/NF drops to -5.75 eV compared with that of Pd/NiMoO₄/NF (-5.25 eV) and Pd/NF (-4.87 eV). The downshift of d-band center is favor of the desorption of carbonyl intermediates (such as glycolate acid) and can promote the timely renewal of the active site, therefore enhances the selectivity of the target products as well as the long-term stability. Although the d-band center of the NiMoO₄/NF is the lowest, it has no obvious EG oxidation activity at low potential due to the absence of active Pd sites.

Figure S11. (a) XPS valence band spectrum of NiMoO₄/NF.

Figure 3. (e) XPS valence band spectra of Pd/NiMoO₄/NF and (f) Pd/NF.

(2) OCPs experiments. **Figure 5a** and **Figure S37** shows the open circuit potential (OCP) measurement results, which can be applied to show the influences of organic adsorbates on the inner Helmholtz layer. Upon adding 1 M EG, the OCP significantly decreases for Pd/NiMoO₄/NF ($\Delta = 0.68$ V) compared with Pd/NF ($\Delta = 0.45$ V) and NiMoO₄/NF ($\Delta = 0.09$ V), indicating that EG is more easily adsorbed

on the Pd/NiMoO₄/NF than on Pd/NF and NiMoO₄/NF surface, which agree well with the high electrocatalytic activity of ethylene glycol oxidation.

Figure S37. OCPs of NiMoO₄/NF in 1 M NaOH solution before and after EG was added.

Figure 5. (a) Open circuit potentials (OCPs) of Pd/NiMoO₄/NF and Pd/NF in 1 M NaOH solution before and after EG was added.

(3) As shown in Figure 5f and Figure S40 and S41, a distinct CO oxidation peak appears for Pd/NF and NiMoO₄/NF in the first anodic scan at the peak potentials at 1.090 V vs RHE and 0.785 V vs RHE (**Figure S40 and S41**). After introducing NiMoO₄, the peak potentials of CO oxidation decrease to a lowered potential 1.026 V vs RHE (**Figure 5f**) for Pd/NiMoO₄/NF, which indicates that CO is weakly adsorbed on the surface and can be easily oxidative-removed on account of the downshift of the d-band center of Pd, eventually contributing to the observed superior EGOR stability.

Figure 5. (f) CO stripping experiments of Pd/NiMoO₄/NF.

Figure S40. CO stripping experiments of Pd/NF.

Figure S41. CO stripping experiments of NiMoO₄/NF.

(4) Details of DFT calculation.

We have employed the Vienna Ab Initio Package (VASP)^{1,2} to perform all the density functional theory (DFT) calculations within the generalized gradient approximation (GGA) using the PBE³ formulation. We have chosen the projected augmented wave (PAW) potentials^{4,5} to describe the ionic cores and take valence electrons into account using a plane wave basis set with a kinetic energy cutoff of 400 eV. Partial occupancies of the Kohn–Sham orbitals were allowed using the Gaussian smearing method and a width of 0.05 eV. The electronic energy was considered self-consistent when the energy change was smaller than 10^{-5} eV. A geometry optimization

was considered convergent when the force change was smaller than 0.02 eV/Å. Grimme's DFT-D3 methodology⁶ was used to describe the dispersion interactions. The equilibrium lattice constant of FCC Pd unit cell was optimized to be $a=3.886$ Å. We then used it to construct a Pd (111) surface model (model 1) with p ($3 \times 2\sqrt{3}$) periodicity in the x and y directions and 4 atomic layers in the z direction separated by a vacuum layer in the depth of 15 Å in order to separate the surface slab from its periodic duplicates. Model 1 comprises of 48 Pd atoms. During structural optimizations, a $2 \times 2 \times 1$ k-point grid in the Brillouin zone was used for k-point sampling, and the bottom two atomic layers were fixed while the top two were allowed to relax.

The equilibrium lattice constants of monoclinic NiMoO₄ unit cell were optimized to be $a=9.468$ Å, $b=8.650$ Å, $c=7.576$ Å, $\alpha=90^\circ$, $\beta=114.2^\circ$, $\gamma=90^\circ$. We then use it to construct a NiMoO₄ ($\bar{3}12$) surface model with p (2×1) periodicity in the x and y directions and 3 stoichiometric layers in the z direction separated by a vacuum layer in the depth of 15 Å in order to separate the surface slab from its periodic duplicates. Model 2 was built by one Pd₁₀ cluster residing onto this NiMoO₄ ($\bar{3}12$) surface. Model 2 comprises of 10 Pd, 24 Ni, 24 Mo and 96 O atoms. During structural optimizations, a $1 \times 2 \times 1$ k-point grid in the Brillouin zone was used for k-point sampling, and the bottom two stoichiometric layers were fixed while the rest were allowed to relax.

The adsorption energy (E_{ads}) of adsorbate A was defined as:

$$E_{\text{ads}} = E_{A/\text{surf}} - E_{\text{surf}} - E_{A(\text{g})}$$

where $E_{A/\text{surf}}$, E_{surf} and $E_{A(\text{g})}$ are the energy of adsorbate A adsorbed on the surface, the energy of clean surface, and the energy of isolated A molecule in a cubic periodic box with a side length of 20 Å and a $1 \times 1 \times 1$ Monkhorst-Pack k-point grid for Brillouin zone sampling, respectively.

The free energy of a gas phase molecule or an adsorbate on the surface was calculated by the equation $G = E + \text{ZPE} - TS$, where E is the total energy, ZPE is the zero-point energy, T is the temperature in kelvin (298.15 K is set here), and S is the entropy.

References

1. Kresse, G.; Furthmüller, J. Efficiency of Ab-Initio Total Energy Calculations for Metals and Semiconductors Using a Plane-Wave Basis Set. *Comput. Mater. Sci.* **1996**, *6*, 15–50.
2. Kresse, G.; Furthmüller, J. Efficient Iterative Schemes for Ab Initio Total-Energy Calculations Using a Plane-Wave Basis Set. *Phys. Rev. B.* **1996**, *54*, 11169–11186.
3. Perdew, J. P.; Burke, K.; Ernzerhof, M. Generalized Gradient Approximation Made Simple. *Phys. Rev. Lett.* **1996**, *77*, 3865–3868.
4. Kresse, G.; Joubert, D. From Ultrasoft Pseudopotentials to the Projector Augmented-Wave Method. *Phys. Rev. B.* **1999**, *59*, 1758-1775.
5. Blochl, P. E. Projector Augmented-Wave Method. *Phys. Rev. B.* **1994**, *50*, 17953–17979.
6. Grimme, S.; Antony, J.; Ehrlich, S.; Krieg, H. J. *Chem. Phys.* **2010**, *132*, 154104.

Reviewer #3: The manuscript submitted by Shi et al reports an interesting electrocatalysts Pd/NiMoO₄/NF for the production of sodium glycolate under ambient conditions. This electrocatalysts shows excellent selectivity and long-term stability. There are a few concerns this reviewer has for this work to be acceptable by Nature Communications, as detailed below.

Response: Thank you very much for taking the time to review my manuscript and for your appreciation to this manuscript. All questions have been responded point by point after careful consideration, please review again.

1. In the introduction, the authors mentioned the importance of sodium glycolate, however no relevant literature was cited to support their claim. For instance, what is the global production of sodium glycolate per year and what is its market price per metric ton?

Response: Many thanks for the comment. The global production of sodium glycolate reaches around 0.1 million tons per year and the market price of sodium glycolate is around 3.0 thousand US dollars. Related information and reference (Ref. 1 and Ref. 2) have been added in the revised manuscript and marked yellow.

2. The authors indicated that a low Pd content (3.5%) was used to prepare this electrocatalyst, however it was not clear what this 3.5% is referring to, weight percentage or mole percentage? And what is the denominator in the calculation? The total mass of Pd/NiMoO₄/NF, or just Pd/NiMoO₄? It is probably to report the mass loading of Pd per geometric area of the electrode. In addition, why 3.5%? A detailed optimization of the Pd loading should be performed and reported.

Response: Many thanks for your kind suggestion.

(1) A low Pd content (3.5 wt.%) is referring to the weight percentage of the total mass of Pd/NiMoO₄/NF. 3.5 wt.% has been used in the revised manuscript for the Pd content to avoid any misleading.

(2) The mass loading of Pd is 1 mg per geometric area (1 mg cm⁻²) of the electrode, which is obviously lower than previously reported references.

(3) Following your suggestion, control samples with lower or higher Pd content have been synthesized and corresponding experiments have been carried out. The control

precursors were prepared by a similar procedure with varied Na_2PdCl_4 concentration. The precise Pd loading amount has been analyzed by inductively coupled plasma mass spectroscopy (ICP-OES), which are 1.7% and 6.9% for $\text{Pd}_{1.7\%}/\text{NiMoO}_4/\text{NF}$ and $\text{Pd}_{6.9\%}/\text{NiMoO}_4/\text{NF}$. As shown in **Figure S16a**, the XRD patterns indicate the successful synthesis of the $\text{Pd}_{1.7\%}/\text{NiMoO}_4/\text{NF}$ and $\text{Pd}_{6.9\%}/\text{NiMoO}_4/\text{NF}$ samples. Noticeably, $\text{Pd}/\text{NiMoO}_4/\text{NF}$ with 3.5 wt.% Pd content shows significantly higher current density than $\text{Pd}_{1.7\%}/\text{NiMoO}_4/\text{NF}$ and $\text{Pd}_{6.9\%}/\text{NiMoO}_4/\text{NF}$ for electrocatalytic ethylene glycol oxidation (**Figure S16b**).

(4) Furthermore, the reason for the much enhanced electrocatalytic activity has been investigated by measuring the adsorption of the OH^- and EG species of these electrocatalysts. As shown in the CV curves (**S17a**), $\text{Pd}/\text{NiMoO}_4/\text{NF}$ exhibits stronger OH^- adsorption bands at significantly lower onset potential (0.5 V vs. RHE and 1.0 vs. RHE) than $\text{Pd}_{1.7\%}/\text{NiMoO}_4/\text{NF}$ and $\text{Pd}_{6.9\%}/\text{NiMoO}_4/\text{NF}$. Besides, upon adding 1 M EG, the OCP is significantly decreased for $\text{Pd}/\text{NiMoO}_4/\text{NF}$ ($\Delta = 0.68$ V) compared with $\text{Pd}_{1.7\%}/\text{NiMoO}_4/\text{NF}$ ($\Delta = 0.52$ V) and $\text{Pd}_{6.9\%}/\text{NiMoO}_4/\text{NF}$ ($\Delta = 0.65$ V), indicating the much enhanced EG adsorption of $\text{Pd}/\text{NiMoO}_4/\text{NF}$ (S17b). The above results indicate that the moderate Pd content is contributed to the adsorption of OH^- and EG and thus enhance the oxidation performance of EG.

The above information has been added in the revised manuscript and supporting information.

Figure S16. (a) XRD patterns of $\text{Pd}_{1.7\%}/\text{NiMoO}_4/\text{NF}$, $\text{Pd}/\text{NiMoO}_4/\text{NF}$ and $\text{Pd}_{6.9\%}/\text{NiMoO}_4/\text{NF}$. (b) LSV curves of obtained electrocatalysts for ethylene glycol anodic oxidation.

Figure S17. (a) CV curves of Pd/NiMoO₄/NF, Pd_{1.7%}/NiMoO₄/NF and Pd_{6.9%}/NiMoO₄/NF in 1 M NaOH. (b) OCPs of Pd/NiMoO₄/NF, Pd_{1.7%}/NiMoO₄/NF and Pd_{6.9%}/NiMoO₄/NF in 1 M NaOH solution with or without EG addition.

3. The choice of NiMoO₄ as the support for the deposition of Pd is not well justified in the text. What is the rationale for using NiMoO₄, instead of NiOx and MoOx? Will the latter two materials show similar or inferior activity compared to NiMoO₄ once Pd is also loaded on them?

Response: Thanks very much for the kind suggestion. Following your suggestion, control samples NiO/NF, Pd/NiO/NF, MoO₃/NF and Pd/MoO₃/NF have been synthesized and corresponding experiments have been carried out. The control precursors were prepared by a similar procedure but without the addition of Ni(NO₃)₂·6H₂O or Na₂MoO₄·2H₂O, which were annealed in Ar atmosphere at the temperature of 450 °C for 2 h in a tubular furnace to obtain the contrast samples. As shown in **Figure S18a**, the XRD patterns of these materials indicate the successful formation of the NiO/NF, Pd/NiO/NF, MoO₃/NF and Pd/MoO₃/NF. Noticeably, the peak current density of Pd/NiMoO₄/NF is significantly higher than those of Pd/NiO/NF and Pd/MoO₃/NF for electrocatalytic ethylene glycol oxidation (**Figure S18b**). Furthermore, the adsorption of the OH⁻ and EG species were determined by the CV and OCP measurements. As shown in **Figure 5e** and **S19a**, Pd/NiMoO₄/NF exhibits OH⁻ adsorption bands at significantly lower onset potential (0.5 V vs RHE) than Pd/NiO/NF (0.65 V vs RHE) and Pd/MoO₃/NF (no obvious adsorption bands). Besides, upon adding 1 M EG, the OCP of Pd/NiMoO₄/NF is significantly decreased ($\Delta = 0.68$ V)

compared with Pd/NiO/NF ($\Delta = 0.65$ V) and Pd/MoO₃/NF ($\Delta = 0.49$ V) (**Figure 5a and S19b**). The above results indicate that the choice of NiMoO₄ as the support for the deposition of Pd is contributed to the adsorption of OH⁻ and EG and thus enhance the oxidation performance of EG.

Figure S18. (a) XRD patterns of NiO/NF, Pd/NiO/NF, MoO₃/NF and Pd/MoO₃/NF. (b) LSV curves of obtained electrocatalysts for ethylene glycol anodic oxidation.

Figure S19. (a) CV curves of Pd/NiMoO₄/NF, Pd/NiO/NF and Pd/MoO₃/NF in 1 M NaOH. (b) OCPs of Pd/NiMoO₄/NF, Pd/NiO/NF and Pd/MoO₃/NF in 1 M NaOH solution with or without EG addition.

4. The detailed mechanism was not investigated in this work. The selectivity (not over oxidation) is critical for the success of this reaction. There are primarily two steps involved, hydrogen abstraction and hydroxide addition, during the alcohol transformation to carboxylate. It is not clear both steps take place on Pd, or NiMoO₄ is somehow also involved. Is the selectivity potential dependent? Based on Figure 4a, the maximum catalytic current appears beyond 1.2 V RHE, however the authors only

conducted electrolysis till 1.1 V. Will increasing applied potential lead to more C-C bond cleaved products? Does the overall oxidation mechanism change when the applied potential is more positive?

Response: Thank you very much for the kind suggestion.

(1) To distinguish the active sites for the hydrogen abstraction and hydroxide addition step, charge density of the Pd and NiMoO₄ sites have been calculated by DFT. As shown in **Figure S10**, the calculation results indicate that 1.63 electrons transfer from the Pd to the NiMoO₄ substrate, which suggests that the Pd is positively charged and the NiMoO₄ substrate is negatively charged. Moreover, the calculated adsorption energies of OH⁻ on the surfaces of Pd/NiMoO₄ are 0.20, 0.26 and 0.28 eV at Pd, Ni and Mo adsorption sites, respectively (**Figure S46**). Therefore, the positively charged Pd is favor of the attraction of anions OH⁻ and the subsequent hydrogen abstraction of EG.

Figure S10. Charge density difference, where the blue and yellow isosurface denote the electron depletion and accumulation on Pd/NiMoO₄.

Figure S46. (a) Calculated adsorption energies of OH on the surfaces of Pd/NiMoO₄ at different adsorption sites. (b) Pd adsorption site. (c) Ni adsorption site. (d) Mo adsorption site.

(2) As shown in **Figure S47**, the calculated adsorption energies of EG on Pd (111) and Pd/NiMoO₄ have also been calculated which are -0.57 and -0.98 eV, respectively, indicating that the EG is more easily adsorbed on the Pd/NiMoO₄ rather than samples without NiMoO₄ support, in accordance with the results of OCP measurements. Besides, the calculated adsorption energies of EG on the surfaces of Pd/NiMoO₄ are -0.98, -0.53 and -0.36 eV at Pd, Ni and Mo adsorption sites, respectively (**Figure S48**), indicating that the EG is more favorable for adsorption at Pd site. Furthermore, as shown in **Figure S28**, the main products of EGOR catalyzed by Pd/NF is sodium glycolate, however, the NiMoO₄/NF has no obvious EG oxidation activity at low potential (**Figure 4a**). The above results indicate that Pd also serves as the active sites for the abstraction of hydrogen.

Figure S47. Calculated adsorption energies of EG on the surfaces of Pd (111) and Pd/NiMoO₄.

Figure S48. (a) Calculated adsorption energies of EG on the surfaces of Pd/NiMoO₄ at different adsorption sites. (b) Pd adsorption site. (c) Ni adsorption site. (d) Mo adsorption site.

(3) Following your suggestion, a long-term chronoamperometric test using Pd/NiMoO₄/NF as the electrocatalyst was carried out at 1.2 V and 1.3 V vs. RHE to investigate the potential dependence of the EG oxidation products. As shown in **Figure 4c**, the faradaic efficiency of 76.0% and 75.8% were obtained at 1.2 V and 1.3 V vs. RHE, respectively, which is much lower than that obtained at 1.1 V and 1.0 V vs. RHE, due to the C-C bond cleavage at more positive potentials (**Figure**

S26). The above results imply the change of the overall oxidation mechanism.

The above results and discussions have been added in the revised manuscript and supporting information.

Figure 4. (c) Faradaic efficiencies (FEs) of Pd/NiMoO₄/NF for sodium glycolate production in 2 h electrolysis at varied potentials.

Figure S26. ¹H NMR spectra of the electrolyte after 2 h anodic ethylene glycol oxidation on Pd/NiMoO₄/NF electrode at varied potentials.

5. Based on Figure 3e and 3f, it is not clear how the authors determine the d-band center. More explanations are needed.

Response: Thank you very much for the kind suggestion. Following your suggestion, the binding energy values were referenced using the C 1 s peak position at 284.8 eV. The d-band centers were evaluated by applying the following formula¹. (*J. Appl. Phys.* **2014**, *115* (12), 124301.

Besides, we further carried out DFT calculations to determine the d-band center of

Pd/NF and Pd/NiMoO₄/NF. As shown in **Figure S12**, the d-band center of Pd/NF and Pd/NiMoO₄/NF are calculated for -1.83 eV and -2.45 eV, respectively, which agree well the results of the surface valance band photoemission spectroscopy (**Figure 3e-f**). Related information has been added in the revised manuscript and marked yellow.

Figure S12. PDOS (d-band) of Pd and Pd/NiMoO₄.

6. Finally, the text should be thoroughly polished. There are quite a few typos or grammatical errors. For instance, line 18 on page 12, “...an electrocatalyst Pd/NiMoO₄/NF with lowered rather low Pd loading (3.5%) ...” reads very odd.

Response: Many thanks for your kind suggestion. The above sentence has been revised to “an electrocatalyst Pd/NiMoO₄/NF with rather low Pd loading (3.5%)”. We have also carefully checked the whole manuscript to correct these grammatical problems.

REVIEWERS' COMMENTS

Reviewer #1 (Remarks to the Author):

The authors have well addressed the comments. In my opinion, the revised manuscript can be accepted on Nat. Commun. in its current form.

Reviewer #2 (Remarks to the Author):

The authors have added required experiments and calculations. The quality of the paper has improved noticeably. I would suggest the paper to be accept.

Just there is a typo in the response letter.

"In particular, the free-energy change of the RDS on

Pd/NiMoO₄ (0.86 eV) is much lower than 0.58 eV of Pd..." I guest the value should be reversed, right?

The author can correct it. No further review is needed from my side.

Reviewer 1: The authors have well addressed the comments. In my opinion, the revised manuscript can be accepted on Nat. Commun. in its current form.

Response: Thank you for your positive recommendation.

Reviewer 2: The quality of the paper has improved noticeably. I would suggest the paper to be accept. Just there is a typo in the response letter.

"In particular, the free-energy change of the RDS on Pd/NiMoO₄ (0.86 eV) is much lower than 0.58 eV of Pd..." I guest the value should be reversed, right? The author can correct it. No further review is needed from my side.

Response: Thank you for your kind recommendation. The above sentence has been revised to "In particular, the free-energy change of the RDS on Pd/NiMoO₄ (0.58 eV) is much lower than 0.86 eV of Pd, suggesting that the Pd/NiMoO₄ catalysts could adsorb OH at a relatively lower onset potential, which is in accordance with the results of CV." Related information has been added in the revised manuscript.